# CLUSTERING DATA WITH GEOMETRIC MODULARITY

## ABSTRACT

Clustering data is a fundamental problem in unsupervised learning with a range of applications in the natural and social sciences. This wide applicability has led to the development of dozens of clustering algorithms. Broadly, these algorithms can be divided as being (i) parametric, e.g. $k$-means, where the centers are parameters and $k$ a hyperparameter, and (ii) non-parametric, e.g. DB-SCAN (Ester et al., 1996), which has hyperparameters, but otherwise only uses density to find clusters. An attractive feature of DB-SCAN is not needing to know the number of clusters (usually unknown in practice) in advance. In this work, we propose a new measure of cluster quality, called *geometric modularity* and show how it can be used to improve popular algorithms such as DB-SCAN. Through experiments on a wide-range of datasets we show that using geometric modularity yields a parameter-free DB-SCAN-based approach with better output quality than other parameter-free density based clustering approaches such as OPTICS and DPC. Interestingly, our experiments also show that *geometric modularity* tracks a *supervised* measure called *adjusted mutual information* well, despite using no label information. Finally, we also provide a theoretical justification of the use of this measure by considering an *idealized* model for well-clusterable data.

## 1 INTRODUCTION

Clustering algorithms are a foundational machine learning tool that find applicability in a wide range of applied fields such as biology, econometrics, forensics, network sciences, political sciences, etc. to extract structured information from amorphous data (Xu & Wunsch, 2005; Xu & Tian, 2015). The goal of clustering algorithms is to divide data into groups, called clusters, in a way that elements in the same cluster are similar to each other while elements in different clusters are dissimilar. Clustering algorithms are widely used in practice for their ability to unveil hidden structures in seemingly unstructured data that can lead to significant insights and discoveries. As a consequence, the design of efficient clustering algorithms has been extensively studied in several areas of computer science and statistics in the past decades.

A key advantage of clustering compared to other machine learning techniques is that clustering algorithms are *unsupervised*, i.e. they do not depend on labeled data that indicates to which cluster a given input belongs. Their ability to learn patterns directly from untagged data is particularly important in settings where gathering labeled data is a challenge, for cost, privacy, or scarcity reasons (e.g., medical diagnosis, etc.). A further application of clustering methods is the potential ability to find structure in data where human experts had hitherto not foreseen any structure; this is particularly relevant for medical and scientific discoveries. To summarize, one of the main reason behind the practical success of clustering lies in its ability to extract information from datasets on which we have very scarce or no *label* information.

However clustering algorithms are very sensitive to hyper-parameters that need to be tuned in a data-dependent manner by an experienced user. Clustering algorithms come in many flavors; we focus on the well-known DB-SCAN algorithm (Ester et al., 1996), a widely-used heuristic, and part of standard machine learning toolboxes (e.g., scikit-learn), that is used for identifying patterns in points embedded in a metric space (Alg. 1). DB-SCAN is a density-based algorithm and it is non-parametric in the sense that the data density determines the number of clusters etc., which need not be known in advance. A key hyper-parameter however for DB-SCAN is the radius,[1] which determines which

---

[1] There is a second hyper-parameter minPoints; but it interacts with the radius and usually has a smaller effect. In `sklearn` it is by default set to 5. We also use the default and find that it is not particularly sensitive.

elements are considered *core*, i.e., points that have many points within this radius, and the connectivity between points. If the radius is set to be too small, DB-SCAN classifies all points as outliers. On the other hand, if the radius is too big then all data will be grouped in a single cluster. In fact, by varying this hyperparameter it is possible to generate several candidate clusterings at different granularities. The *radius* in DB-SCAN is typically treated as a hyper-parameter. One of the best-known and widely used method to automatically select this hyperparameter is OPTICS (Ankerst et al., 1999), which can be seen as a hierarchical clustering version of DB-SCAN. At a high-level, OPTICS ranks the points based on how far they are from dense regions of the space. OPTICS then defines the clusters by identifying key break points in the ordering (when two consecutive points have very different neighbor densities). We also compare with Density Peak Clustering (DPC) which identifies local peaks in terms of density and assigns each point to the nearest local peak to form clusters.

---

**Algorithm 1:** DB-SCAN (simplified)

---

1 **Input:** Data: $\mathbf{x}_1, \ldots, \mathbf{x}_n$, distance $d(\cdot, \cdot) \to \mathbb{R}^+$; Hyper-Parameters: $\varepsilon$, minPoints
2 Set $\mathbf{x}_i$ to be core if $\varepsilon$-ball around $\mathbf{x}_i$ has at least minPoints points
3 Add a directed edge from $\mathbf{x}_i$ to $\mathbf{x}_j$ whenever $\mathbf{x}_i$ is core and $d(\mathbf{x}_i, \mathbf{x}_j) \leq \varepsilon$
4 Say $\mathbf{x}_j$ reachable from $\mathbf{x}_i$ if there is a directed path, denote $\mathbf{x}_i \to \mathbf{x}_j$
5 Say $\mathbf{x}_i \sim \mathbf{x}_j$ if there exists $\mathbf{x}_k$ with $\mathbf{x}_k \to \mathbf{x}_i$ and $\mathbf{x}_k \to \mathbf{x}_j$
6 Obtain the transitive closure of $\sim$ and output equivalence classes as clusters

---

**Our Results**: This work focuses on improving on previous non-parametric methods with a focus on DB-SCAN. We start by designing a method to autotune, in an unsupervised fashion, the radius parameter, denoted by $\varepsilon$, of DB-SCAN. Then we show how to improve the results further via a local search routine.

In the supervised setting a natural way to set the hyperparameter $\varepsilon$ would be to select the value for which the clusters output by DB-SCAN have the highest adjusted mutual information score (AMI)[2] with respect to the target clusters induced by the labeled data. AMI is a measure of similarity between two candidate clusterings (for a formal definition of AMI refer to Section 4). As we are in the unsupervised setting, we introduce a new quality measure for clusterings (i.e., partitionings of the space), called *Geometric modularity*, that evaluates the quality of a clustering (without the knowledge of the target clusters).

Our approach is based on a generalization of the notion of modularity (Newman, 2006) to vector data. A striking property of our new notion of modularity is its ability of mimicking the behavior of the adjusted mutual information (AMI) without having access to the ground truth. In fact, we can show empirically that our new notion of modularity, despite being an *unsupervised measure* tracks the *supervised measure* AMI, incredibly well on datasets where the underlying clustering is known. For an example of this phenomena refer to Figure 1. In particular, it is remarkable that these two measures achieve the maximum for very similar values of $\varepsilon$ (the scales are different, though irrelevant when choosing the mode). In Appendix A.1 we show that this behavior is, to a large extent, replicated on a wide range of datasets.

This metric allows us to develop a novel approach to tune $\varepsilon$ creating an essentially hyperparameter-free clustering algorithm. Concretely, we simply pick the value of $\varepsilon$ for which DB-SCAN maximizes the Geometric modularity and outputs the clusters hence obtained. The details of the algorithm are provided in Section 2 after we formally define Geometric modularity.

From a theoretical perspective, we show that our technique is able to recover the underlying clustering in a well-separated clustering model related to that of Arora et al. (2018b). In particular, we show that using our Geometric modularity metric, it is possible to auto-tune the parameter of DB-SCAN to recover the underlying clustering structure.

From an experimental perspective, we compare our method to OPTICS, the main approach for choosing the radius parameter of DB-SCAN, the density peak clustering algorithm (DPC) (Rodriguez & Laio, 2014), and to a *hypothetical* algorithm that picks the hyper-parameter using the AMI score which is a supervised quantity. OPTICS seeks to *order* points in a dataset in terms of reachability distance from previously considered points and uses steep changes in this distance to automatically

---

[2]Or any other suitable supervised accuracy measure.

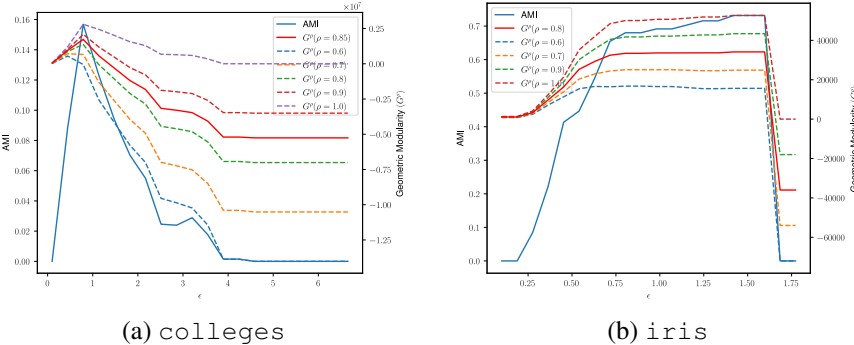

(a) `colleges`                    (b) `iris`

Figure 1: We show the AMI and the Geometric modularity score for different clusterings obtained by using DB-SCAN with different $\varepsilon$ to cluster the `colleges` and `iris` datasets. Geometric modularity has a *resolution parameter* $\rho$, which we show how to pick automatically in a data-dependent way in Section 4. These plots also show that this is not a sensitive hyper-parameter as the curves are broadly similar across a range of values of $\rho$.

identify new clusters. Although, this algorithm still has a hyperparameter, $\xi$, to indicate the steepness of the change, in libraries it is often implemented with defaults that yield good results on certain datasets. DPC is based on the idea that "cluster centers" are local maxima in terms of their density and far from other centers. We show that our approach significantly outperforms OPTICS and for hyper-parameter tuning of DB-SCAN. On all our datasets DPC did not produce meaningful results. In addition, we also show that Geometric modularity is able to track closely the AMI score on multiple datasets, proving its effectiveness in detecting a good clustering.

**Local Search**: We moreover show experimentally that the results obtained via DB-SCAN, namely the results obtained by running DB-SCAN with parameter $\varepsilon$ chosen so as to maximize Geometric modularity, can be further improved by running a simple local search heuristic seeded with DB-SCAN clusters. Concretely, the local search algorithm iteratively moves a point from one cluster to another if it increases the overall Geometric modularity of the clustering. We show that seeded with the DB-SCAN clusters (which are useful to identify the overall structure), the local search algorithm often outperforms DB-SCAN in terms of AMI.

**Additional related work**: There has been a lot of attention to compute the optimal parameters for DB-SCAN algorithm. Unfortunately most of previous works consider a setting that significantly departs from ours. Bergstra et al. (2011); Karami & Johansson (2014); Zhang et al. (2022)) assume that the algorithm has access to some set of already labeled data elements (i.e.: the semi-supervised setting) and their approaches cannot be applied to the fully unsupervised setting we consider here. Other approaches, such as Liu et al. (2007); Mitra & Nandy (2011), introduce another parameter that replaces the parameter $\varepsilon$. Namely, both approaches run DB-SCAN with a specific value $\varepsilon_p$ for each point $p$ of the input. The main issue is that in both cases, the value $\varepsilon_p$ is derived from the distances from $p$ to its $k$-nearest neighbors, where $k$ is a new parameter. The approaches thus replace the parameter $\varepsilon$ with a new parameter, $k$. Finally, none of the previous work (Bergstra et al. (2011); Karami & Johansson (2014); Zhang et al. (2022); Liu et al. (2007); Mitra & Nandy (2011); Lai et al. (2019); Zhou & Gao (2014)) prove theoretical guarantees of the resulting algorithm. So our work if the first method with good experimental result and theoretical guarantees. Some work focuses on improving the running time or scalability, see e.g.: Esfandiari et al. (2021); Jiang et al. (2020).

From a theoretical perspective not much work has been done for DB-SCAN. Recently few papers (Sriperumbudur & Steinwart (2012); Jiang (2017); Steinwart et al. (2017); Jang & Jiang (2019)) study the statistical properties of DB-SCAN and few of its variants although our paper is the first paper in which it is shown that DB-SCAN can recover "clusterable" instances. Despite the practical significance of DB-SCAN, its statistical properties have only been explored recently. Such analyses make use of recent developments in topological data analysis to show that DB-SCAN estimates the connected components of a level-set of the underlying density.

## 2 GEOMETRIC MODULARITY

In our setting we receive as input a set of points and a distance (or dissimilarity) function between them. In this setting, an ideal cluster corresponds to a sets of points that are "unusually" close in comparison to the global distance structure of the input at hand. To capture this intuition we present the notion of *geometric modularity* and we show that this notion aligns very well with the target ground-truth clusters, and so enables hyper-parameter tuning for DB-SCAN. Geometric modularity is defined for vector-valued inputs and is inspired by modularity introduced by Newman (2006) in the context of detecting communities in networks. The main intuition behind our notion is to assign a score to each point that is proportional to the difference between its "average" distance to the rest of the points in the instance and the "average" distance to the rest of the points in its cluster. Arenas et al. (2008) considered a more general version of the modularity function to capture different resolutions at which cluster structures may appear. Subsequently, Newman (2016) established the equivalence between modularity maximization and maximum likelihood methods for community detection in block-models in this more general case. We will also allow the additional resolution parameter, which we denote by $\rho$ in our definition of geometric modularity. Although in principle this introduces an extra hyper-parameter, in Section 4 we show how this can also be tuned automatically from data.

Formally, let $\mathbf{x}_1, \ldots, \mathbf{x}_n$ be $n$ points in $\mathbb{R}^m$ and let $d(\cdot, \cdot)$ denote some distance function (smaller distance indicates greater similarity). We do not require $d(\cdot, \cdot)$ to be a metric. In Section 2.1, we show how geometric modularity can be computed efficiently when $d(\cdot, \cdot)$ is the squared *Euclidean* distance. In our experiments, we also use squared Euclidean distance. For any $\mathbf{x}_i$, let $D_i = \sum_j d(\mathbf{x}_i, \mathbf{x}_j)$. And let $Z = \sum_i D_i$. Let $c_1, \ldots, c_n$ be an assignment of points to a cluster, where each $c_i$ is an integer between 1 and $n$ and it represents the cluster to which $\mathbf{x}_i$ is assigned. Let $\delta_{c_i c_j}$ be equal to 1 if $c_i = c_j$ and 0 otherwise. Then, we define the geometric modularity ($G^\rho$) of a clustering as:

$$G^\rho = \sum_{i,j} \left( \rho \cdot \frac{D_i D_j}{Z} - d(\mathbf{x}_i, \mathbf{x}_j) \right) \delta_{c_i c_j}$$

In general, higher values of $G^\rho$ indicate higher quality of the underlying clustering. To understand better our new notion, let's consider the contribution of one point to $G^\rho$. Let's first assume that $\rho = 1$, though we will discuss the effect of $\rho$ below when discussing the properties of geometric modularity. When $\rho = 1$, we will ignore the cumbersome superscript. In particular, we can rewrite $G = \sum_i G_i$ where $G_i = D_i \sum_j \left( \frac{D_j}{Z} - \frac{d(\mathbf{x}_i, \mathbf{x}_j)}{D_i} \right) \delta_{c_i c_j}$. We note that the score of a point is directly proportional to its distance to the rest of the points in the instance. Furthermore two points benefit more from being in the same cluster if their distances to the rest of the points is larger. In particular, the contribution to the score that one gets by placing $\mathbf{x}_i$ and $\mathbf{x}_j$ in the same cluster is $\frac{D_j}{Z} - \frac{d(\mathbf{x}_i, \mathbf{x}_j)}{D_i}$, where $\frac{D_j}{Z}$ captures the "average" distance of point $j$ to the remaining points in the instance and $\frac{d(\mathbf{x}_i, \mathbf{x}_j)}{D_i}$ captures the local distance between point $j$ to point $i$. This intuitively captures the fact that for a set of points to be in the same cluster their distance has to be "unusually" close in comparison with the set of distances in the instance – connecting with the philosophy of clustering: Points that are close should be in the same clusters, while points that are far should be in separate clusters.

**Basic Properties of Geometric Modularity**: Let's now turn our attention to some basic properties of geometric modularity. First note that if all points are clustered together (e.g. $c_i = 1$ for all $i$), then $G^\rho = (\rho - 1)Z$; in particular, when $\rho = 1$, then we have $G = 0$. This captures the fact that there is no gain in clustering the instance into a single cluster. Second, if we cluster every point in a singleton cluster (e.g. $c_i = i$ for all $i$) then the total score of the instance is $G^\rho = \rho \sum_i \frac{D_i^2}{Z}$. This also allows us to see the effect of $\rho$. In general, if $\rho \ll 1$, we will prefer smaller clusters, while if $\rho \gg 1$, we will prefer larger (and fewer) clusters. In Section 4, we give a method to tune $\rho$ from data, typically in the range $[0.5, 1]$. Essentially, we can count the number of clusters obtained by picking $\varepsilon$ to maximize the geometric modularity for different values of $\rho$. We find that the number of clusters is relatively *stable* as a function of $\rho$ and pick $\rho$ to be in the interval with the most stability in this regard.

### 2.1 COMPUTATIONAL COMPLEXITY

In this section, we show how to efficiently compute the modularity of a given clustering of a given dataset. Let $X = \{\mathbf{x}_1, \ldots, \mathbf{x}_n\}$ denote a dataset of $n$ points in an $m$-dimensional Euclidean space,

and for points $\mathbf{x}, \mathbf{y}$, the dissimilarity function is the squared Euclidean distance $d(\mathbf{x}, \mathbf{y}) = \|\mathbf{x} - \mathbf{y}\|^2$. Let $\mathcal{C} = \{C_1, \ldots, C_k\}$ be a partition of the $X$. We now explain how the modularity of $\mathcal{C}$ can be computed in time linear in $n$ and $m$.[3] Concretely, we wish to compute

$$G^\rho = \sum_{i,j} \left( \rho \frac{D_i D_j}{Z} - d(\mathbf{x}_i, \mathbf{x}_j) \right) \delta_{c_i c_j}$$

where $c_i$ indicates which cluster of $\mathcal{C}$ contains $\mathbf{x}_i$.

1. Let $\mu$ be the mean of $X$ and let $\Sigma_\mu = \sum_{i=1}^n \|\mathbf{x}_i - \mu\|_2^2$. We have that for any point $\mathbf{p}$ in $\mathbb{R}^m$, $\sum_{i=1}^n \|\mathbf{x}_i - \mathbf{p}\|_2^2 = \Sigma_\mu + n\|\mu - \mathbf{p}\|_2^2$. Hence, computing $D_i = \Sigma_\mu + n\|\mu - \mathbf{x}_i\|_2^2$ for all $i \in [n]$ can be done in linear time by first computing $\mu$ and then $\Sigma_\mu$, which can easily be done in linear time. It follows that $Z$ can be computed in linear time.

2. Similarly, we let $\mu_{C_\ell}$ and $\Sigma_{\mu_{C_\ell}}$ be respectively the mean of cluster $C_\ell$ and the sum of distances squared of the points in $C_\ell$ to the mean of cluster $C_\ell$. Again, computing the means of all the clusters can be done in linear time. Moreover, by an argument similar to the above one, $\sum_{i,j|c_i=c_j} \|\mathbf{x}_i - \mathbf{x}_j\|_2^2$ can be rewritten as $\sum_{i=1}^n (\Sigma_{\mu_{c_i}} + |c_i| \cdot \|\mathbf{x}_i - \mu_{c_i}\|_2^2)$ and computed in linear time.

3. Finally, for each cluster $C_\ell$, we let $Z_\ell = \sum_{i|c_i=C_\ell} D_i$ and we rewrite the sum $\sum_{ij|c_i=c_j} D_i D_j$ as $\sum_{i|c_i=C_\ell} D_i Z_\ell = Z_\ell^2$. Since computing $Z_\ell$ for all $\ell \in [k]$ can be done in linear time, we have that $\sum_{ij|c_i=c_j} D_i D_j = \sum_{\ell=1}^k Z_\ell^2$ can be computed in linear time.

Therefore, by rewriting $G^\rho$ as

$$G^\rho = \frac{\rho}{Z} \sum_{\ell=1}^k Z_\ell^2 - \sum_{i=1}^n (\Sigma_{\mu_{c_i}} + |c_i| \cdot \|\mathbf{x}_i - \mu_{c_i}\|_2^2)$$

and since $Z$, $\sum_{\ell=1}^k Z_\ell^2$, and $\sum_{i=1}^n (\Sigma_{\mu_{c_i}} + |c_i| \cdot \|\mathbf{x}_i - \mu_{c_i}\|_2^2)$ can be computed in linear time, $G^\rho$ can be computed in linear time.

## 3 THEORETICAL INSIGHTS

In this section, we formally justify the use of Geometric modularity by showing that it can determine ground-truth clusters in inputs that exhibit a clear clustering. Missing proofs appear in Appendix C.

We focus on a beyond-worst-case analysis of the behavior of the Geometric modularity objective; indeed, worst-case analysis for clustering is frequently not useful as real-life instances are often very far from being worst-case. Beyond worst-case analysis has been a successful research direction in the context of clustering: such analyses have shown that $k$-means, combined with low-dimensional projections, e.g., Principal component analysis (PCA), or t-SNE, indeed unveil the cluster structure of a dataset when it exists. For example, Arora et al. (2018a) have analyzed the performance of the popular t-SNE embedding on instances exhibiting an underlying "spherical" and "well-separated" clustering, and Kumar & Kannan (2010) have shown that $k$-means and PCA allows to recover "well-separated" clustering. For graph modularity, Cohen-Addad et al. (2020) have shown that the Louvain heuristic recovers the planted clusters in graphs drawn from the stochastic block model, a distribution of graphs exhibiting a clear cluster structure. On the other hand, for several popular heuristics, such as DB-SCAN, nothing similar has been shown so far.

In this section, we prove two results on what we call 'regular' instances that consist of 'well-separated' clusters $\{C_1, \ldots, C_k\}$:

1. Theorem 3.2: The partition that maximizes *geometric modularity* is the one corresponding to the well-separated clusters, namely $\{C_1, \ldots, C_k\}$.

2. Theorem 3.6: The DB-SCAN algorithm, tuned using geometric modularity, outputs the well-separated clusters, namely $\{C_1, \ldots, C_k\}$.

---

[3]Note that a naïve algorithm to compute modularity would require quadratic time in $n$

For regular and well-separated instances, the output of DB-SCAN already maximizes geometric modularity, hence the additional local search step is not needed (though is not harmful!). So this analysis holds equally for the algorithm with the local search procedure. Our experiments show that on real-world data the local search step has benefits.

**Beyond-worst-case instances**: We first provide a formal definition of the beyond-worst-case instances we consider. This is a variant of that used by Arora et al. (2018a) for the t-SNE algorithm. Unlike them, our notion does not require that the planted clusters be spherical, but requires some basic regularity. We show that instances that don't satisfy such a regularity notion may indeed exhibit several underlying ground-truth clusterings; and so no theorems as the above two can be derived.

**Definition 3.1** (Well-separated, regular data). *Let $X = \{\mathbf{x}_1, \mathbf{x}_2, \ldots, \mathbf{x}_n\} \subset \mathbb{R}^m$ be clusterable data with $\mathcal{C} = \{C_1, C_2, \ldots, C_k\}$ defining the individual clusters such that for each $i \in [k]$, $|C_i| \geq 0.1(n/k)$. We say that $X$ is $\eta$-regular and $\gamma$-well-separated if for some $b > 0$, we have:*

1. *$\eta$-Regular: For any point $\mathbf{x}_i$, $\eta^{-1}nb^2 \leq \sum_{j=1}^n \|\mathbf{x}_j - \mathbf{x}_i\|_2^2 \leq \eta nb^2$.*

2. *$\gamma$-Well-Separated Clustering: For any $\ell, \ell' \in [k]$, $(\ell \neq \ell')$, $i \in C_\ell$ and $j \in C_{\ell'}$, we have $\|\mathbf{x}_i - \mathbf{x}_j\| \geq (1 + \gamma)b$; and for any $i \in C_\ell$, we have $|\{j \in C_\ell \setminus \{i\} : \|\mathbf{x}_i - \mathbf{x}_j\| \leq b/4\}| \geq 0.51|C_\ell|$.*

**Structural properties of beyond-worst-case instances**: We show that the partition that maximizes the geometric modularity objective agrees with the planted clusters; and that regularity and well-clusterability assumptions are needed for this to hold. It is worth commenting briefly on the parameters, $\eta, \gamma$ and $b$; $b$ represents the scale of the instance; $\eta$ we think of as a constant very close to 1, representing high regularity, and $\gamma$ is a constant greater than 1 ensuring minimum distance between points in different clusters. In particular, this means that the interval $(\eta^3/4, (1+\gamma)^2\eta^{-3})$ is non-empty and includes 1; for relatively large values of $\gamma$ this interval allows some flexibility with respect to the choice of $\rho$ in the theorem below.

**Theorem 3.2.** *For any $\eta$-regular and $\gamma$-well-separated instance with clusters $\mathcal{C} = \{C_1, C_2, \ldots, C_k\}$, we have that for any resolution parameter value $\rho$ such that $\eta^3/4 < \rho < (1+\gamma)^2\eta^{-3}$, $\mathcal{C}$ is the partition that maximizes geometric modularity with resolution $\rho$.*

To prove Theorem 3.2, we use the following two lemmas, whose proofs can be found in Appendix C.

**Lemma 3.3.** *Let $\rho < (1+\gamma)^2\eta^{-3}$. Consider a clustering $\mathcal{S} = \{S_1, \ldots, S_{k'}\}$. For any $i \in [k']$, $j \in [k]$, define $S^{i,j} := S_i \cap C_j$. If there exist $i, j, j'$ such that $j \neq j'$ and $S^{i,j} \neq \emptyset$ and $S^{i,j'} \neq \emptyset$, then the clustering $\mathcal{S}^* := \mathcal{S} - \{S_i\} \cup \bigcup_{\ell=1}^k \{S^{i,\ell}\}$ has higher geometric modularity than $\mathcal{S}$.*

**Lemma 3.4.** *Let $\rho > \eta^3/4$. Consider a clustering $\mathcal{S} = \{S_1, \ldots, S_{k'}\}$. If there exist $i, j, \ell$ such that $S_i \subset C_\ell$ and $S_j \subset C_\ell$. Then the clustering $\mathcal{S}^* := \mathcal{S} - \{S_i\} - \{S_j\} \cup \{S_i \cup S_j\}$ has higher geometric modularity than $\mathcal{S}$.*

The above two lemmas imply Theorem 3.2 (see Appendix C). We now show that our regularity condition is indeed required since otherwise multiple "ground-truth" clusterings can coexist in the same instance, at different resolution levels. In Figure 2 we show that the regularity condition is needed. In the case where there is high irregularity in the data, there may be two well-separated clusterings co-existing in the input.

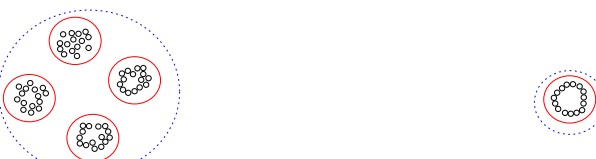

Figure 2: Two different well-separated clusterings represented in red continuous line and blue dotted line. The resolution parameter allows to identify each.

**Geometric modularity to self-parameterized DB-SCAN**: We consider the geometric modularity objective for any $\rho$ such that $\eta^3/4 < \rho < (1+\gamma)^2\eta^{-3}$. Our proof of Theorem 3.6 relies on the following lemma, whose proof can be found in Appendix C.

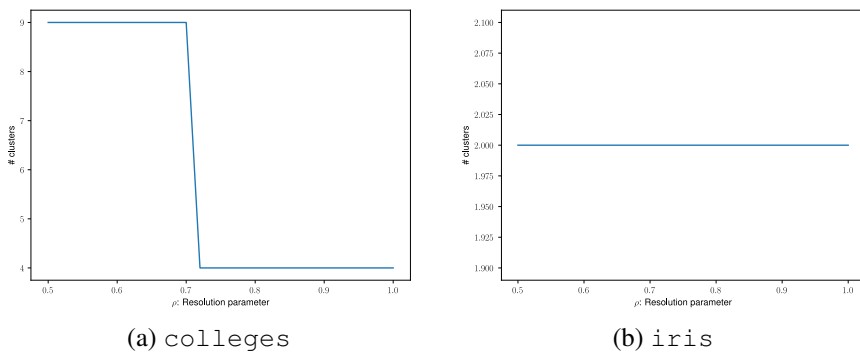

(a) `colleges`    (b) `iris`

Figure 3: A plot of #clusters vs $\rho$ on the datasets `colleges` and `iris`.

**Lemma 3.5.** *Let $\mathcal{C} = \{C_1, \ldots, C_k\}$ denote the clusters of an $\eta$-regular and $\gamma$-well-separated clustering instance. Let $\varepsilon \in [b, (1 + \gamma)b)$, and $\mathrm{minPoints} \geq .51|C_\ell|$ for all $\ell$. Then, the clusters output by* DB-SCAN *with parameters $\varepsilon$ and $\mathrm{minPoints}$ is a permutation of $\mathcal{C}$.*

Equipped with the above lemma, the proof of Theorem 3.6 follows immediately (see Appendix C).

**Theorem 3.6.** *Consider any $\eta$-regular and $\gamma$-well-separated clustering instance with clusters $\mathcal{C} = \{C_1, C_2, \ldots, C_k\}$. Suppose* DB-SCAN *is run with $\mathrm{minPoints} \geq 0.51|C_\ell|$ for all $\ell \in [k]$ and varying values of $\varepsilon$ to generate candidate clusters. The candidate clustering that maximizes geometric modularity, $G^\rho$ for $\rho \in (\eta^3/4, (1 + \gamma)^2/\eta^3)$ is a permutation of $\{C_1, \ldots, C_k\}$.*

**Discussion**: We have not attempted to optimize any of the parameters in our results, and our main goal is to show that in *idealized* settings our algorithm will work correctly. We use this theoretical grounding to evaluate our algorithms empirically and observe that it compares favorably to existing methods.

## 4 Results and Discussion

In this section, we explain the setup for our experimental results, the methodology, the choice of datasets and a summary of the results. Further plots and and results appear in Appendices A and B.

### 4.1 Methodology

Our algorithm is outlined in Section 2. From an implementation perspective there are two key choices. First, we need to pick the resolution parameter, $\rho$, in the definition of the function $G^\rho$. For any fixed $\rho$, we can run DB-SCAN with various values of $\varepsilon$ and pick the clustering that maximizes the value of $G^\rho$. Note that for a fixed $\varepsilon$, this clustering is unique (when DB-SCAN is implemented deterministically; otherwise, we can smooth all our results over a small number of runs). We keep track of the number of clusters produced and denote that by $k(\rho)$. We repeat this for different values of $\rho$ and create a plot of $k(\rho)$ vs. $\rho$ for $\rho$ in some range. In our experiments we picked $\rho \in [0.5, 1]$.

DB-SCAN can have a large number of outliers, depending on the value of $\varepsilon$. We don't count the outliers as clusters with a singleton element. When outliers are ignored, we find that in most cases, the number of clusters is not very sensitive to the hyper-parameter $\rho$. This is shown on the datasets `colleges` and `iris` in Figure 3. This phenomenon is fairly common across a wide range of datasets (cf. Appendix A). We pick $\rho$ to be any value in the longest sub-interval for which the number of clusters remains fixed. Although, we do not need to do this in our experiments, we find that as $k(\rho)$ is (generally) a decreasing function of $\rho$, we can first perform *isostonic regression* to smooth the curve before selecting $\rho$ (Barlow & Brunk, 1972).

**Clustering Algorithm:** Once the value of $\rho$ is picked, we pick $\varepsilon$ for which the output clustering of DB-SCAN maximizes $G^\rho$. We noticed that performing local search on the resulting cluster to allow *swaps* that increase $G^\rho$ mildly improves the performance. We report performance metrics for our algorithm both with and without local search.

| dataset | optics | | | | dbscan-mod (LS) | | | | dbscan-ami (hypothetical) | | | |
|---|---|---|---|---|---|---|---|---|---|---|---|---|
| | ami | prec. | recall | f1 | ami | prec. | recall | f1 | ami | prec. | recall | f1 |
| biomed | 0.06 | 0.90 | 0.03 | 0.05 | **0.34** | 0.83 | 0.40 | **0.54** | 0.16 | 0.84 | 0.37 | 0.51 |
| colleges | 0.02 | 0.82 | 0.00 | 0.00 | **0.27** | 0.53 | 0.58 | **0.56** | 0.16 | 0.58 | 0.54 | **0.56** |
| diabetes | 0.00 | 0.63 | 0.00 | 0.00 | **0.07** | 0.69 | 0.19 | 0.30 | 0.04 | 0.70 | 0.31 | **0.43** |
| euca | 0.09 | 0.28 | 0.05 | 0.08 | **0.17** | 0.29 | 0.25 | 0.27 | 0.15 | 0.23 | 0.86 | **0.36** |
| gesture | 0.01 | 0.88 | 0.00 | 0.00 | **0.08** | 0.26 | 0.44 | **0.32** | 0.01 | 0.25 | 0.45 | **0.32** |
| har | 0.01 | 1.00 | 0.00 | 0.00 | **0.51** | 0.34 | 0.88 | **0.49** | 0.21 | 0.34 | 0.47 | 0.40 |
| iris | 0.14 | 1.00 | 0.05 | 0.09 | 0.64 | 0.58 | 0.95 | 0.72 | **0.73** | 0.60 | 1.00 | **0.75** |
| libras | 0.30 | 0.80 | 0.10 | 0.17 | **0.46** | 0.23 | 0.45 | **0.31** | 0.34 | 0.14 | 0.39 | 0.21 |
| magic | 0.00 | 0.81 | 0.00 | 0.00 | **0.08** | 0.61 | 0.36 | 0.45 | 0.06 | 0.61 | 0.85 | **0.71** |
| mice | 0.41 | 1.00 | 0.05 | 0.09 | 0.26 | 0.23 | 0.26 | **0.25** | **0.48** | 0.59 | 0.11 | 0.19 |
| musk | 0.03 | 0.96 | 0.00 | 0.00 | 0.04 | 0.77 | 0.34 | **0.48** | **0.09** | 0.82 | 0.18 | 0.30 |
| olivetti | **0.45** | 0.99 | 0.25 | **0.40** | 0.44 | 0.11 | 0.55 | 0.18 | 0.42 | 0.69 | 0.24 | 0.35 |
| pendigits | 0.04 | 1.00 | 0.00 | 0.00 | 0.57 | 0.34 | 0.74 | 0.47 | **0.63** | 0.57 | 0.62 | **0.59** |
| skdigits | 0.06 | 1.00 | 0.01 | 0.02 | **0.64** | 0.57 | 0.59 | **0.58** | 0.43 | 0.84 | 0.31 | 0.46 |
| wine | 0.06 | 1.00 | 0.01 | 0.02 | **0.68** | 0.97 | 0.61 | **0.75** | 0.34 | 0.59 | 0.54 | 0.56 |

Table 1: Comparison dbscan-mod (LS), OPTICS and dbscan-ami in terms of AMI and F-score

**Competitor Auto-tuning DB-SCAN Algorithms:** There are relatively few automated methods that tune the hyper-parameter $\varepsilon$ for DB-SCAN. The most famous of these is OPTICS (Ankerst et al., 1999), which is implemented in the `sklearn` library (Pedregosa et al., 2011) and is widely used. We also report results obtained using OPTICS. We find that when data is not sufficiently low-dimensional, OPTICS fails quite badly and tends to have a large number of outliers. We also ran experiments with Density Peak Clustering (DPC) (Rodriguez & Laio, 2014); using a publicly available implementation,[4] we found that this algorithm always returned a single cluster as the output. Since the main point of our work is to automatically tune DB-SCAN parameters, we believe it is fair to compare competitor algorithms without extensive hyper-parameter tuning. Nevertheless, we did try different values of the hyper-parameter $\xi$ for OPTICS; a range of choices don't change the fundamental picture. We report some of these results in Appendix A.3.

We also compare ourselves to a hypothetical algorithm which uses *adjusted mutual information* (AMI) with the ground-truth clusters as a means of picking the hyper-parameter $\varepsilon$. Obviously, this cannot be implemented in an *unsupervised* fashion as this requires supervision, i.e. access to the ground-truth labels. Nevertheless, we find that our *fully unsupervised* algorithm performs almost as well (and often even better) than this hypothetical supervised algorithm that cannot be implemented in a fully unsupervised environment.

**Datasets:** We use several small to medium scale datasets from the UCI machine learning repository Dua & Graff (2017) (hence publicly available). The datasets that we chose are those for which ground-truth clusters are available; in certain cases these are essentially multi-class classification problems, each class forming a single cluster. The list of datasets we use is in Table 1.

## 4.2 SUMMARY OF RESULTS

We report the following metrics for the clusters output by the algorithms. We let $\mathbf{c} = (c_1, \ldots, c_n)$ be the cluster labels for the points $\mathbf{x}_1, \ldots, \mathbf{x}_n$ output by the clustering algorithm and let $\mathbf{c}^\star = (c_1^\star, \ldots, c_n^\star)$ denote the ground-truth cluster labels. We assume that each of $c_i$ and $c_i^\star$ are in $\{1, \ldots, n\}$.

**Adjusted Mutual Information:** We associate a joint distribution over random variables $(X, Y)$, where $X = c_i, Y = c_i^\star$ with probability $\frac{1}{n}$. Then the *adjusted mutual information* between $\mathbf{c}$ and $\mathbf{c}^\star$, $\text{ami}(\mathbf{c}, \mathbf{c}^\star)$ is defined as $\text{ami}(\mathbf{c}, \mathbf{c}^\star) = \frac{I(X;Y) - \mathbb{E}I(X;Y)}{(H(X)+H(Y))/2 - \mathbb{E}I(X;Y)}$, where $H$ denote the entropy, $I(X;Y)$ denotes the mutual information between $X$ and $Y$, and $\mathbb{E}I(X;Y)$ denotes the *expected* mutual information if $X$ and $Y$ were independent partitions with same the number of clusters and the number of points in each clusters as in the joint case (Vinh et al., 2009).

---

[4]https://github.com/lanbing510/DensityPeakCluster

| dataset | dbscan-mod (LS) | | | | dbscan-mod | | | | dbscan-ami (hypothetical) | | | |
|---|---|---|---|---|---|---|---|---|---|---|---|---|
| | ami | prec. | recall | f1 | ami | prec. | recall | f1 | ami | prec. | recall | f1 |
| biomed | **0.34** | 0.83 | 0.40 | **0.54** | 0.16 | 0.84 | 0.37 | 0.51 | 0.16 | 0.84 | 0.37 | 0.51 |
| colleges | **0.27** | 0.53 | 0.58 | **0.56** | 0.16 | 0.58 | 0.54 | **0.56** | 0.16 | 0.58 | 0.54 | **0.56** |
| diabetes | **0.07** | 0.69 | 0.19 | 0.30 | 0.03 | 0.84 | 0.04 | 0.08 | 0.04 | 0.70 | 0.31 | **0.43** |
| euca | **0.17** | 0.29 | 0.25 | 0.27 | 0.12 | 0.24 | 0.29 | 0.26 | 0.15 | 0.23 | 0.86 | **0.36** |
| gesture | **0.08** | 0.26 | 0.44 | **0.32** | 0.01 | 0.25 | 0.45 | **0.32** | 0.01 | 0.25 | 0.45 | **0.32** |
| har | **0.51** | 0.34 | 0.88 | **0.49** | 0.21 | 0.34 | 0.47 | 0.40 | 0.21 | 0.34 | 0.47 | 0.40 |
| iris | 0.64 | 0.58 | 0.95 | 0.72 | **0.73** | 0.60 | 1.00 | **0.75** | **0.73** | 0.60 | 1.00 | **0.75** |
| libras | **0.46** | 0.23 | 0.45 | **0.31** | 0.33 | 0.38 | 0.15 | 0.21 | 0.34 | 0.14 | 0.39 | 0.21 |
| magic | **0.08** | 0.61 | 0.36 | 0.45 | 0.04 | 0.77 | 0.22 | 0.34 | 0.06 | 0.61 | 0.85 | **0.71** |
| mice | 0.26 | 0.23 | 0.26 | **0.25** | **0.48** | 0.59 | 0.11 | 0.19 | **0.48** | 0.59 | 0.11 | 0.19 |
| musk | 0.04 | 0.77 | 0.34 | **0.48** | 0.07 | 0.78 | 0.29 | 0.43 | **0.09** | 0.82 | 0.18 | 0.30 |
| olivetti | **0.44** | 0.11 | **0.55** | 0.18 | 0.42 | 0.69 | 0.24 | 0.35 | 0.42 | 0.69 | 0.24 | 0.35 |
| pendigits | 0.57 | 0.34 | 0.74 | 0.47 | **0.63** | 0.57 | 0.62 | **0.59** | **0.63** | 0.57 | 0.62 | **0.59** |
| skdigits | **0.64** | 0.57 | 0.59 | **0.58** | 0.43 | 0.84 | 0.31 | 0.46 | 0.43 | 0.84 | 0.31 | 0.46 |
| wine | **0.68** | 0.97 | 0.61 | **0.75** | 0.26 | 1.00 | 0.21 | 0.35 | 0.34 | 0.59 | 0.54 | 0.56 |

Table 2: Comparison dbscan-mod (LS), dbscan-mod and dbscan-ami in terms of AMI and F-score

**Precision, Recall, F1-score:** For two every two points $\mathbf{x}_i$ and $\mathbf{x}_j$, let $y_{ij} = 1$ if they are in the same output cluster and 0 otherwise. We define analogously the ground-truth value $y_{ij}^* = \delta_{c_i^\star, c_j^\star}$, i.e. $y_{ij}^\star = 1$ if $\mathbf{x}_i$ and $\mathbf{x}_j$ are in the same cluster in the ground-truth clustering given by $\mathbf{c}^\star$ and 0 otherwise.

This allows us to define *precision*, *recall* and the F1-score in the standard way. Namely, F1-score $= 2/(\text{precision}^{-1} + \text{recall}^{-1})$ where

$$\text{precision} = \frac{|\{(i,j) \mid y_{ij} = 1 \wedge y_{ij}^\star = 1\}|}{|\{(i,j) \mid y_{ij} = 1\}|}, \quad \text{recall} = \frac{|\{(i,j) \mid y_{ij} = 1 \wedge y_{ij}^\star = 1\}|}{|\{(i,j) \mid y_{ij}^\star = 1\}|}.$$

**Results:** The results are shown in Tables 1 and 2. In Table 1, as can be seen from the columns for OPTICS, as there are a large number of outliers, the precision of the clusters output by OPTICS is pretty high, but the recall is quite low. Likewise, the AMI is quite low. This is because the support of the distribution generated by the clusters is much larger that the distribution generated by the ground-truth clustering. On the other hand, we can also see that while the precision can be somewhat lower for our method, the corresponding recall is quite high. As a result our algorithm consistently outperforms OPTICS both on F1-score and AMI. In fact, our algorithm compares extremely favorably even to the hypothetical algorithm which uses AMI directly to select the hyper-parameter $\varepsilon$. In addition to the metrics in Table 1, we also report the correlation coefficients of the outputs of these algorithms with the ground-truth in Appendix A.4 as suggested by Gösgens et al. (2021).

Perhaps surprisingly it shows that DB-SCAN with geometric modularity and local search, dbcan-mod (LS) performs even better than the hypothetical algorithm in terms of AMI. The fact that this is because of local search is clear from Table 2, where we also report performance of picking an $\varepsilon$ for DB-SCAN using geometric modularity, but without local search. We see that this is still comparable in performance to the hypothetical algorithm and in particular, much better than OPTICS. In Table 5 in Appendix B we report some properties of the datasets as well as the number of clusters and outliers produced by the algorithms.

## CONCLUSION AND FUTURE WORK

We introduce a new measure of cluster quality called Geometric modularity and we show that it can be used to tune hyperparameters of clustering algorithms effectively. Leveraging this new measure we improve over the classic DB-SCAN algorithm and we show both theoretically and experimentally the strength of newly introduced technique. Natural next steps are to leverage this new measure to improve other clustering techniques or to introduce new algorithms to optimize it directly.

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

# A ADDITIONAL EXPERIMENTAL RESULTS

## A.1 FURTHER EXPERIMENTAL PLOTS: AMI AND GEOMETRIC MODULARITY

In this section, we show the plots of the relationship between AMI and Geometric modularity $G_\rho$ for all datasets used in the paper. The X-axis is the radius $\varepsilon$ for DB-SCAN and the Y-axes have the AMI (thick blue) and Geometric modularity ($G_\rho$). The thick red line shows $G_\rho$ for $\rho$ picked as per the method described in Section 4; the other lines show $G_\rho$ for different values of $\rho$ as described in the legend.

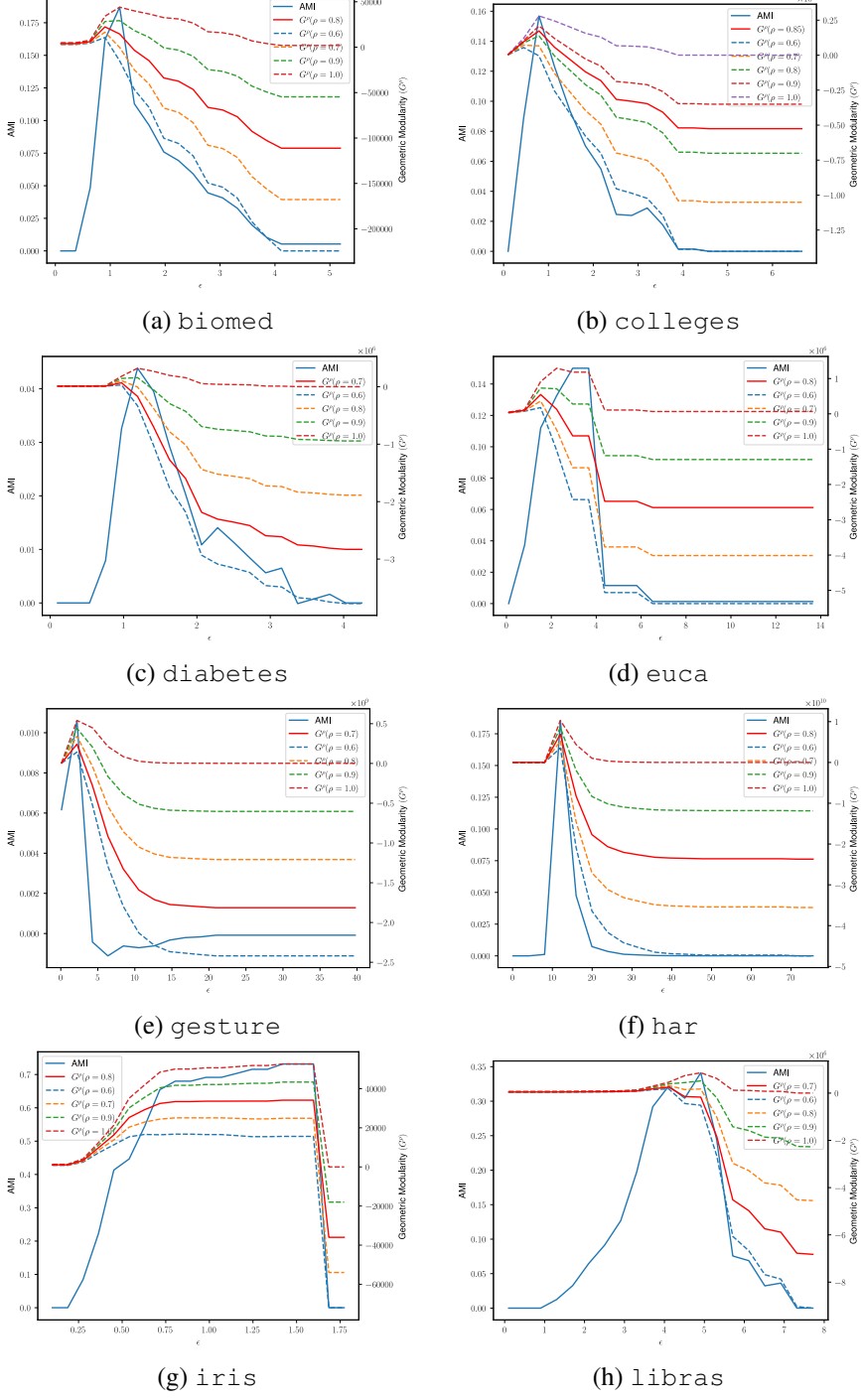

(a) biomed

(b) colleges

(c) diabetes

(d) euca

(e) gesture

(f) har

(g) iris

(h) libras

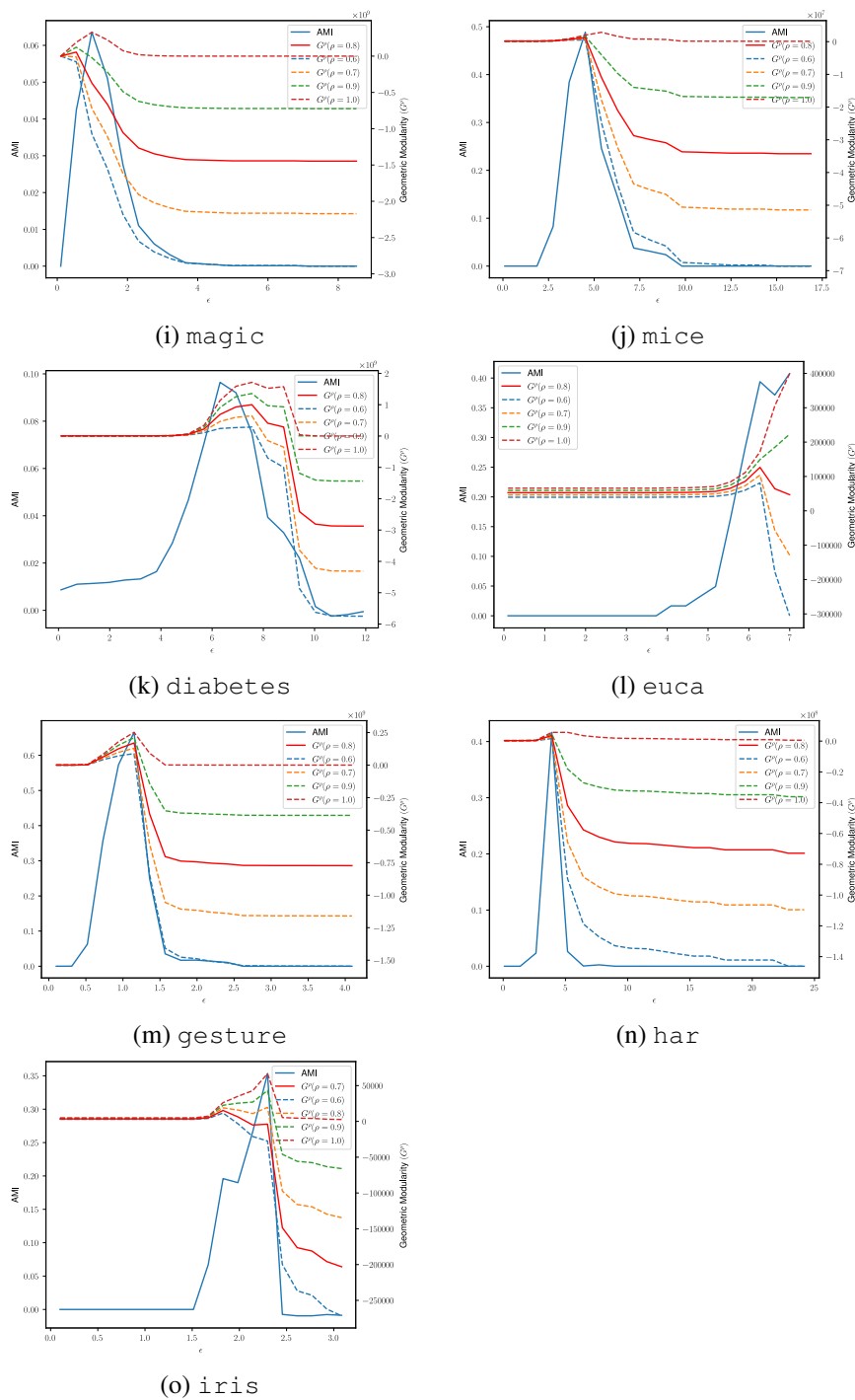

(i) magic

(j) mice

(k) diabetes

(l) euca

(m) gesture

(n) har

(o) iris

## A.2 FURTHER EXPERIMENTAL PLOTS: # CLUSTERS VS RESOLUTION PARAMETER ($\rho$)

We show plots similar to Figure 3 for all datasets used in our paper. We notice that for some datasets the curve is not monotonic. This is because when $\rho$ is small, most points are classified as singletons, and these outliers are not counted as clusters. These plots show that $\rho$ is not a sensitive hyperparameter and can be automatically picked from the data in an easy manner.

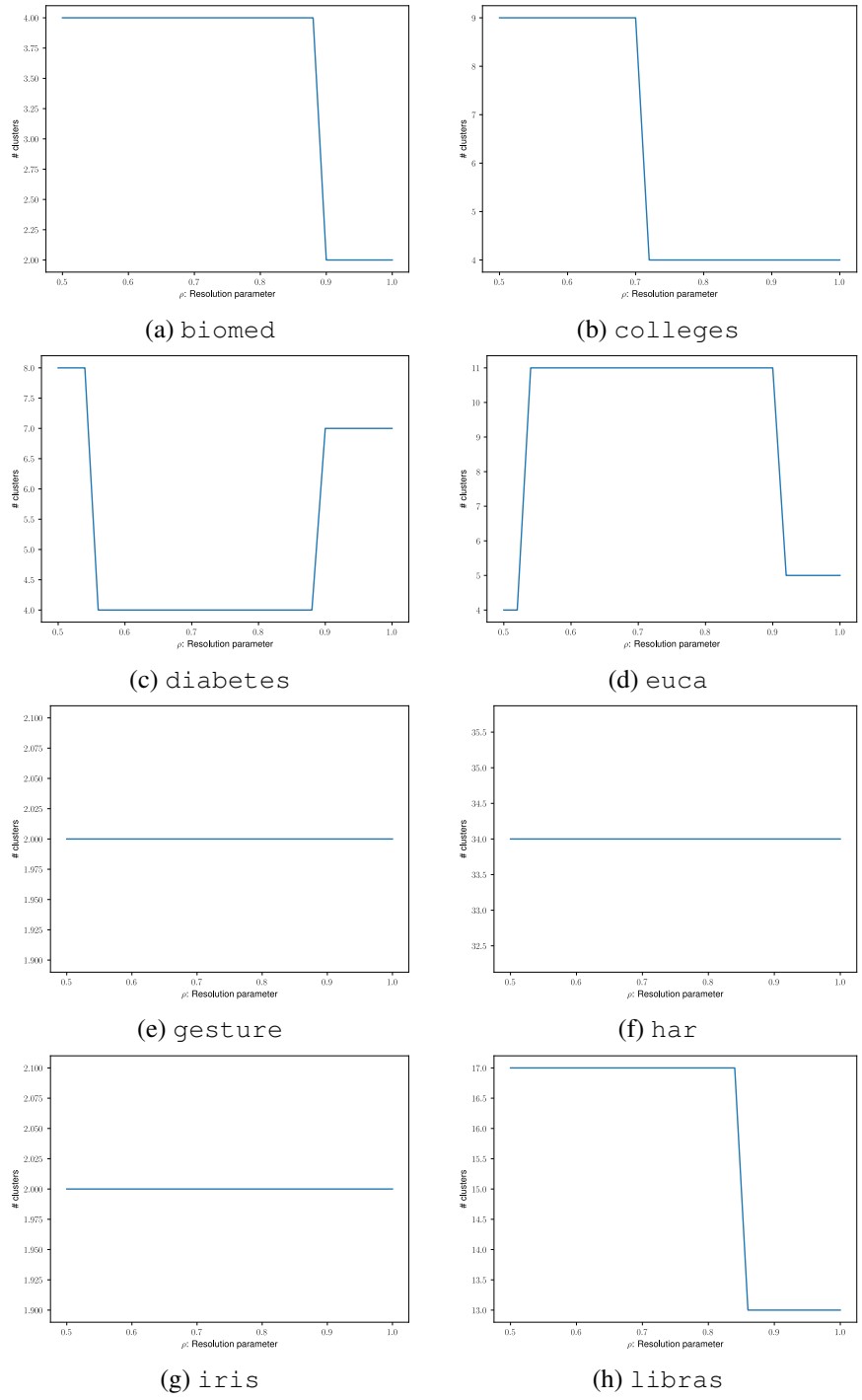

(a) biomed

(b) colleges

(c) diabetes

(d) euca

(e) gesture

(f) har

(g) iris

(h) libras

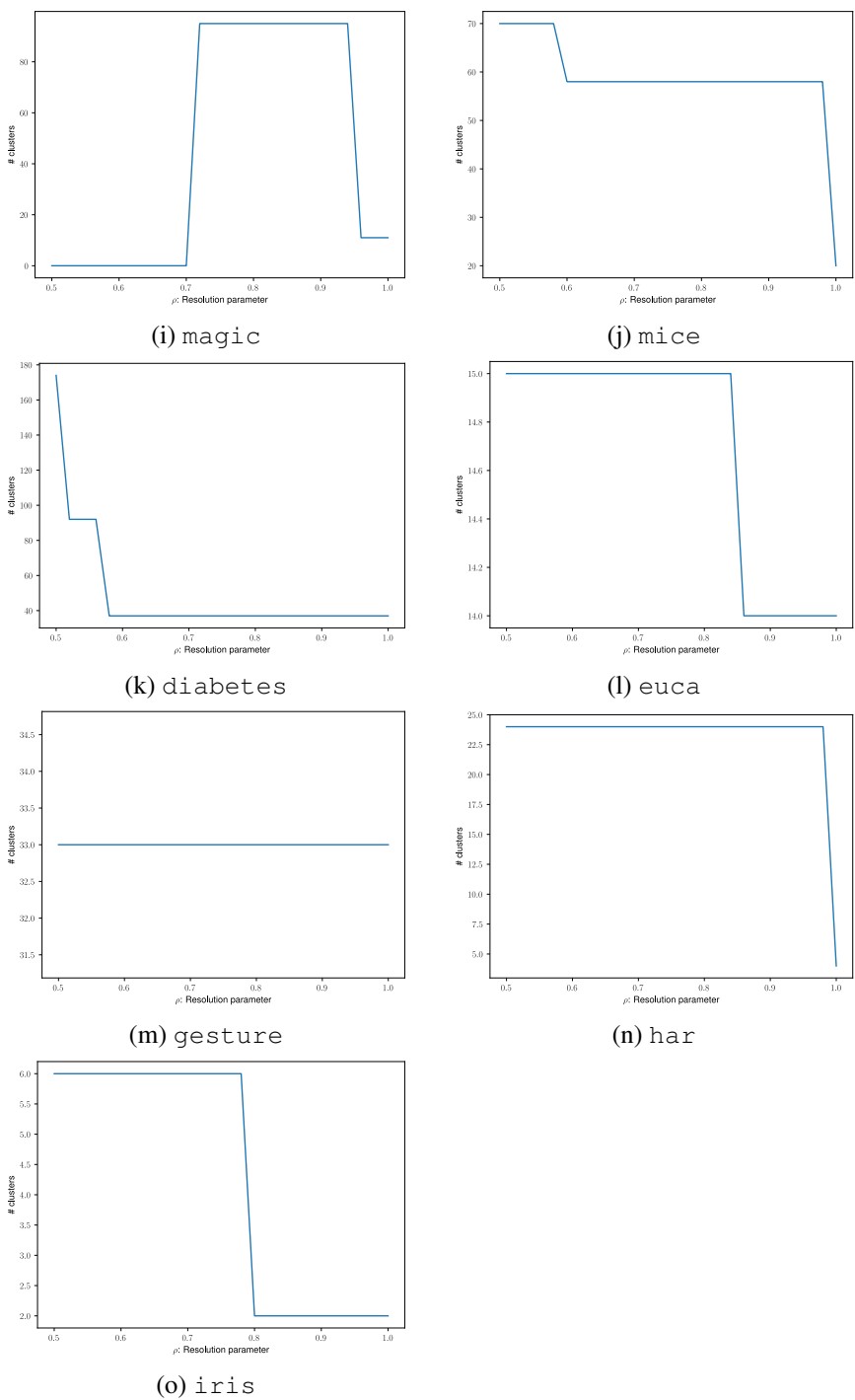

(i) `magic`

(j) `mice`

(k) `diabetes`

(l) `euca`

(m) `gesture`

(n) `har`

(o) `iris`

## A.3 TUNING THE HYPERPARAMETER $\xi$ FOR OPTICS

The hyperparameter $\xi$ in OPTICS controls the *steepness* which is used to define a new cluster; lower values of $\xi$ result in coarser clusters. The default value in `sklearn` is set to 0.05. We report the AMI, precision, recall and F1 score for 5 values of $\xi$ on all datasets. We see that moderate changes to $\xi$ do not fundamentally alter the picture that OPTICS is not as effective as our approach when it comes to automatically tuning DB-SCAN hyperparameters.

| dataset | $\xi$ | ami | prec | recall | f1 |
|---------|-------|-----|------|--------|-----|
| biomed | 0.0125 | 0.08 | 0.88 | 0.04 | 0.07 |
| | 0.025 | 0.07 | 0.88 | 0.04 | 0.07 |
| | 0.05 | 0.06 | 0.90 | 0.03 | 0.05 |
| | 0.1 | 0.05 | 0.91 | 0.03 | 0.06 |
| | 0.2 | 0.01 | 1.00 | 0.00 | 0.00 |
| colleges | 0.0125 | 0.04 | 0.79 | 0.00 | 0.00 |
| | 0.025 | 0.03 | 0.80 | 0.00 | 0.00 |
| | 0.05 | 0.02 | 0.82 | 0.00 | 0.00 |
| | 0.1 | 0.01 | 0.91 | 0.00 | 0.00 |
| | 0.2 | 0.01 | 0.88 | 0.00 | 0.00 |
| diabetes | 0.0125 | 0.01 | 0.67 | 0.00 | 0.00 |
| | 0.025 | 0.01 | 0.69 | 0.00 | 0.00 |
| | 0.05 | 0.00 | 0.63 | 0.00 | 0.00 |
| | 0.01 | 0.00 | 0.52 | 0.00 | 0.00 |
| | 0.02 | 0.00 | 0.71 | 0.00 | 0.00 |
| euca | 0.0125 | 0.11 | 0.53 | 0.02 | 0.03 |
| | 0.025 | 0.11 | 0.52 | 0.02 | 0.03 |
| | 0.05 | 0.09 | 0.28 | 0.05 | 0.08 |
| | 0.01 | 0.04 | 0.27 | 0.03 | 0.05 |
| | 0.02 | 0.02 | 0.23 | 0.02 | 0.03 |
| gesture | 0.0125 | 0.02 | 0.78 | 0.00 | 0.00 |
| | 0.025 | 0.01 | 0.83 | 0.00 | 0.00 |
| | 0.05 | 0.01 | 0.88 | 0.00 | 0.00 |
| | 0.1 | 0.01 | 0.93 | 0.00 | 0.00 |
| | 0.2 | 0.00 | 0.90 | 0.00 | 0.00 |
| har | 0.0125 | 0.04 | 1.0 | 0.00 | 0.00 |
| | 0.025 | 0.03 | 1.0 | 0.00 | 0.00 |
| | 0.05 | 0.01 | 1.0 | 0.00 | 0.00 |
| | 0.1 | 0.00 | 1.0 | 0.00 | 0.00 |
| | 0.2 | 0.00 | 0.17 | 1.0 | 0.29 |
| iris | 0.0125 | 0.20 | 1.0 | 0.05 | 0.10 |
| | 0.025 | 0.20 | 1.0 | 0.06 | 0.12 |
| | 0.05 | 0.14 | 1.0 | 0.05 | 0.09 |
| | 0.1 | 0.08 | 1.0 | 0.03 | 0.06 |
| | 0.2 | 0.72 | 0.59 | 0.99 | 0.74 |
| libras | 0.0125 | 0.39 | 0.67 | 0.15 | 0.25 |
| | 0.025 | 0.37 | 0.76 | 0.14 | 0.23 |
| | 0.05 | 0.33 | 0.82 | 0.11 | 0.20 |
| | 0.1 | 0.25 | 0.93 | 0.08 | 0.15 |
| | 0.2 | 0.11 | 0.90 | 0.03 | 0.05 |

| dataset | $\xi$ | ami | prec | recall | f1 |
|---------|-------|-----|------|--------|-----|
| magic | 0.0125 | 0.01 | 0.83 | 0.00 | 0.00 |
| | 0.025 | 0.01 | 0.83 | 0.00 | 0.00 |
| | 0.05 | 0.00 | 0.81 | 0.00 | 0.00 |
| | 0.1 | 0.00 | 0.84 | 0.00 | 0.00 |
| | 0.2 | 0.00 | 1.00 | 0.00 | 0.00 |
| mice | 0.0125 | 0.44 | 1.0 | 0.05 | 0.10 |
| | 0.025 | 0.42 | 1.0 | 0.05 | 0.10 |
| | 0.05 | 0.40 | 1.0 | 0.05 | 0.10 |
| | 0.1 | 0.39 | 1.0 | 0.05 | 0.10 |
| | 0.2 | 0.23 | 1.0 | 0.03 | 0.06 |
| musk | 0.0125 | 0.04 | 0.94 | 0.00 | 0.00 |
| | 0.025 | 0.04 | 0.94 | 0.00 | 0.00 |
| | 0.05 | 0.03 | 0.96 | 0.00 | 0.00 |
| | 0.01 | 0.02 | 0.98 | 0.00 | 0.00 |
| | 0.02 | 0.01 | 0.99 | 0.00 | 0.00 |
| olivetti | 0.0125 | 0.51 | 0.91 | 0.31 | 0.46 |
| | 0.025 | 0.49 | 0.94 | 0.30 | 0.45 |
| | 0.05 | 0.45 | 0.99 | 0.25 | 0.40 |
| | 0.01 | 0.16 | 1.0 | 0.06 | 0.12 |
| | 0.02 | 0.11 | 1.0 | 0.04 | 0.08 |
| pendigits | 0.0125 | 0.11 | 1.0 | 0.00 | 0.00 |
| | 0.025 | 0.09 | 1.0 | 0.00 | 0.00 |
| | 0.05 | 0.04 | 1.0 | 0.00 | 0.00 |
| | 0.1 | 0.02 | 1.0 | 0.00 | 0.00 |
| | 0.2 | 0.00 | 1.0 | 0.00 | 0.00 |
| skdigits | 0.0125 | 0.15 | 1.0 | 0.01 | 0.02 |
| | 0.025 | 0.11 | 1.0 | 0.01 | 0.02 |
| | 0.05 | 0.05 | 1.0 | 0.01 | 0.02 |
| | 0.1 | 0.01 | 1.0 | 0.00 | 0.00 |
| | 0.2 | 0.00 | 0.01 | 1.0 | 0.18 |
| wine | 0.0125 | 0.18 | 1.0 | 0.05 | 0.09 |
| | 0.025 | 0.14 | 1.0 | 0.04 | 0.07 |
| | 0.05 | 0.06 | 1.0 | 0.01 | 0.02 |
| | 0.1 | 0.00 | 0.34 | 1.0 | 0.51 |
| | 0.2 | 0.00 | 0.34 | 1.0 | 0.51 |

Table 3: Performance metrics for different values of $\xi$ in DB-SCAN.

## A.4 Correlation Coefficients between Clusters and the Ground-Truth

Motivated by the inadequacy of the existing metrics such as NMI, AMI, F1-score, etc. Gösgens et al. (2021) suggested also evaluating cluster metrics using the Pearson correlation coefficient between the output clusters and the ground truth. We report these for all datasets for the three methods: optics, dbscan-mod, dbscan-mod-LS.

| dataset | dbscan-mod-LS | dbscan-mod | optics |
|---|---|---|---|
| biomed | **0.35** | 0.34 | 0.09 |
| colleges | 0.24 | **0.28** | 0.22 |
| diabetes | **0.12** | 0.10 | 0.00 |
| euca | **0.09** | 0.03 | 0.03 |
| gesture | **0.05** | 0.04 | 0.01 |
| har | **0.4** | 0.25 | 0.01 |
| iris | 0.58 | **0.63** | 0.18 |
| libras | 0.26 | 0.20 | **0.29** |
| magic | 0.09 | **0.20** | 0.00 |
| mice | 0.13 | **0.22** | 0.20 |
| musk | 0.05 | **0.06** | 0.01 |
| olivetti | 0.21 | 0.40 | **0.50** |
| pendigits | 0.42 | **0.55** | 0.02 |
| skdigits | **0.53** | 0.49 | 0.08 |
| wine | **0.69** | 0.39 | 0.09 |

Table 4: Correlation coefficient between output clusters and ground-truth clusters for the different algorithms.

# B DATASETS AND SOME ADDITIONAL RESULTS

The table below shows some general properties of the datasets we used as well as some additional experimental results. In particular, for each clustering algorithm we report the number of clusters produced as well as the number of outliers. It is clear that several of these algorithms produce lots of outliers for some datasets and that the local search procedure improves this. This also explains why we occasionally get worse precision, but better recall, and as a result a better $F1$ score.

| dataset | parameters | | | optics | | | dbcsan-mod | | dbscan-mod (LS) | | dbscan-ami | |
|---|---|---|---|---|---|---|---|---|---|---|---|---|
| | $n$ | $k$ | dim | # cl | # ol | $\rho$ | # cl | # ol | # cl | # ol | # cl | # ol |
| biomed | 209 | 3 | 7 | 5 | 152 | 0.80 | 3 | 70 | 6 | 1 | 3 | 70 |
| colleges | 1161 | 4 | 13 | 20 | 1019 | 0.85 | 5 | 386 | 5 | 0 | 5 | 386 |
| diabetes | 768 | 2 | 8 | 10 | 704 | 0.70 | 7 | 585 | 11 | 2 | 2 | 386 |
| euca | 736 | 5 | 14 | 19 | 358 | 0.80 | 8 | 19 | 9 | 0 | 3 | 2 |
| gesture | 9873 | 5 | 32 | 61 | 9381 | 0.70 | 4 | 3457 | 46 | 3 | 4 | 3457 |
| har | 10299 | 6 | 561 | 22 | 10126 | 0.80 | 25 | 3949 | 5 | 1 | 25 | 3949 |
| iris | 150 | 3 | 4 | 5 | 108 | 0.80 | 2 | 0 | 2 | 0 | 2 | 0 |
| libras | 360 | 15 | 90 | 19 | 223 | 0.70 | 17 | 178 | 13 | 1 | 13 | 112 |
| magic | 19020 | 2 | 10 | 155 | 17955 | 0.80 | 98 | 10459 | 9 | 0 | 7 | 2438 |
| mice | 1080 | 8 | 77 | 84 | 321 | 0.80 | 59 | 172 | 14 | 1 | 59 | 172 |
| musk | 6598 | 2 | 166 | 323 | 4035 | 0.80 | 27 | 61 | 5 | 0 | 102 | 499 |
| olivetti | 400 | 40 | 4096 | 20 | 260 | 0.80 | 17 | 262 | 19 | 1 | 17 | 262 |
| pendigits | 10992 | 10 | 16 | 128 | 9991 | 0.80 | 43 | 1314 | 10 | 0 | 43 | 1314 |
| skdigits | 1797 | 10 | 64 | 18 | 1643 | 0.80 | 24 | 807 | 16 | 0 | 24 | 807 |
| wine | 178 | 3 | 13 | 4 | 154 | 0.70 | 5 | 97 | 8 | 2 | 2 | 45 |

Table 5: This table shows for each dataset the number of points $n$, number of clusters $k$, the data dimension dim, as well as the number of clusters (# cl) and number of outliers (# ol) produced by all the algorithms. For our algorithms we also show the resolution parameter $\rho$ that was picked for geometric modularity.

## C    PROOFS OF SECTION 3

In this section, we include the missing proofs from Section 3.

*Proof of Lemma 3.3.* We simply compute the change in modularity going from $\mathcal{S}$ to $\mathcal{S}^*$, i.e., letting $\Delta := G^\rho(\mathcal{S}^*) - G^\rho(\mathcal{S})$, we show that $\Delta > 0$. To do this, it is enough to bound the change in contribution to the modularity objective induced by the points $p \in \bigcup_{j=1}^k S^{i,j} \cup S^{i,\ell}$. (We slightly abuse notation, by having the sets $S_i$ include the indices of points, so the corresponding point will be $\mathbf{x}_p$.) We thus have

$$\Delta \geq \sum_{j \in k} \sum_{\substack{\ell \in k \\ \ell \neq j}} \sum_{p \in S^{i,j}} \sum_{q \in S^{i,\ell}} -\rho \frac{D_p D_q}{Z} + \|\mathbf{x}_p - \mathbf{x}_q\|_2^2$$

We now argue that for each pair $p \in S^{i,j}$ and $q \in S^{i,\ell}$, for $j \neq \ell$, we have that $\|\mathbf{x}_p - \mathbf{x}_q\|_2^2 > \rho D_q D_p / Z$ which combined with the above argument yields the lemma. By our assumption, we have that $\|\mathbf{x}_p - \mathbf{x}_q\|_2^2 > (1+\gamma)^2 b^2$, since $p \in C_j$ and $q \in C_\ell$; and $\eta^{-1} n b^2 < D_p < \eta n b^2$ for all $p$, and so $Z > \eta^{-1} n^2 b^2$. Therefore, $\rho D_q D_p / Z < \rho \eta^3 b^2$ Since $\rho < (1+\gamma)^2 \eta^{-3}$, the lemma follows.    $\square$

*Proof of Lemma 3.4.* We again compute the change in modularity going from $\mathcal{S}$ to $\mathcal{S}^*$, i.e., letting $\Delta := G^\rho(\mathcal{S}^*) - G^\rho(\mathcal{S})$, we show that $\Delta > 0$. To do this, it is enough to bound the change in contribution to the modularity objective induced by the points $p \in S_i \cup S_j$. We thus have

$$\Delta \geq \sum_{p \in S_i} \sum_{q \in S_j} \rho \frac{D_p D_q}{Z} - \|\mathbf{x}_p - \mathbf{x}_q\|_2^2$$

We will show that $\|\mathbf{x}_p - \mathbf{x}_q\|_2^2 < \rho D_q D_p / Z$. By our assumption, we have that for any $p' \in C_\ell$, $|\{q' \in C_\ell \mid \|\mathbf{x}_{p'} - \mathbf{x}_{q'}\|_2 \leq b/4\}| \geq .51 |C_\ell|$ and so for any $p', q' \in C_\ell$

$$\{p'' \in C_\ell \mid \|\mathbf{x}_{p'} - \mathbf{x}_{p''}\|_2 \leq b/4\} \cap \{q' \in C_\ell \mid \|\mathbf{x}_{q'} - \mathbf{x}_{p''}\|_2 \leq b/4\} \neq \emptyset$$

Thus, there is a point that is at distance at most $b/4$ from both $\mathbf{x}_p$ and $\mathbf{x}_q$. Triangle inequality immediately implies $\|\mathbf{x}_p - \mathbf{x}_q\|_2^2 < b^2/4$, Moreover, $\eta^{-1} b^2 < D_p < \eta n b^2$ for all $p$, and so $Z < \eta n^2 b^2$. Therefore, $\rho D_q D_p / Z > \rho b^2 \eta^{-3}$. Since $\rho > \eta^3/4$, the lemma follows.    $\square$

*Proof of Theorem 3.2.* Lemma 3.3 implies that any clustering $\mathcal{S}$ that contains a cluster which has non-empty intersection with two clusters $C_i, C_j$ can be improved into a clustering such that each cluster is a subset of a cluster $C_i$. Next, Lemma 3.4 shows that any clustering $\mathcal{S}$ containing two clusters $S_i, S_j$ that are subsets of a cluster $C_i$ can be improved by merging $S_i$ and $S_j$ into a single cluster. Therefore, the clustering $C_1, \ldots, C_k$ has the highest modularity.    $\square$

*Proof of Lemma 3.5.* Clearly, if $\epsilon < (1+\gamma)b$, then the clusters output by DB-SCAN cannot overlap several clusters since the graph generated does not contain an edge between any pair of points at distance larger than $b$. So we have to show that each $C_\ell \in \mathcal{C}$ is fully contained in a cluster output by Algorithm 1, which then in fact corresponds to a cluster output by Algorithm 1. Furthermore, note that any point that belongs to a cluster $C_\ell$ has at least $.51 |C_\ell|$ neighbors at distance at most $b$ and since minPoints is at least $.51 |C_\ell|$, all the input points are core points.

We next show that for each cluster $C_i$, for any $p, q \in C_i$, either $p$ is reachable from $q$ or $q$ is reachable from $p$ or there exists a core point $p^*$ and both $p$ and $q$ are reachable from $p^*$ and $p^*$ is reachable from both $p$ and $q$. Indeed, by the definition of the well-clusterable clustering, $p$ and $q$ have at least $0.1 |C_\ell|$ common points at distance at most $b$. Since $0.1 |C_\ell| > 0$ by the definition and that all points are core points, there exists a core point that is reachable and can reach both $p$ and $q$. It follows that $p$ and $q$ are in the same cluster output by Algorithm 1.    $\square$

*Proof of Theorem 3.6.* Recall that the algorithm returns the maximum modularity clustering computed. Moreover, Lemma 3.5 implies that the clustering $\{C_1, \ldots, C_k\}$ is computed by the algorithm as one of the candidate clusterings, and Theorem 3.2 states that this the maximum modularity clustering over the whole instance. Therefore, the algorithm will return the clustering $\{C_1, \ldots, C_k\}$ as prescribed desired.    $\square$

