# OpenReview forum: "Clustering with Geometric Modularity"
_ICLR.cc/2024/Conference — Submitted to ICLR 2024_

### Official Review · Reviewer_q8CK · 2023-10-30

**Soundness:** 4 excellent
**Presentation:** 4 excellent
**Contribution:** 4 excellent
**Rating:** 8
**Confidence:** 4

**Summary:**

The authors proposed a new measurement for evaluating the quality of clustering called geometric modularity. Inspired by the metric modularity in network, for each data point geometric modularity calculates the difference between its “average” distance to the all the rest of the points and those in its cluster. Both theoretical and empirical results are provided which shows the effectiveness of the proposed measurement.

**Strengths:**

1. A novel measurement for evaluating clustering quality is proposed.
2. The new measurement is linear in computation.
3. Both theoretical and strong empirical results are provided.

**Weaknesses:**

1. The connection of proposed geometric modularity and density-based clustering is not clearly stated.
2. Some part of experiments are not described clearly enough to reproduce the results.
3. More comparison methods are needed.

**Questions:**

1. The connection of proposed geometric modularity and density-based clustering is not directly explained. From the definition of geometric modularity, it can be used for tuning hyper-parameters for any clustering method (the title also implies this) and is not directly related to DBSCAN. But the authors only evaluated it with DBSCAN. The authors are recommended to explain more about this, why not other clustering methods. Maybe it is due to empirical observations, then it would be nice to give evaluations of using geometric modularity on other clustering algorithms.
2. The method described in 4.1 is not clear enough for the others to reproduce the results. For instance, the searching ranging of \epsilon, granularity of \rho and detailed description of local search. BTW, will the authors open source the code to improve the reproducibility?
3. The authors are recommended to compare with HDBSCAN, which is known to be robust to hyper-parameters. And its implementation is also available in sklearn.

---

> ### Author Response · Authors · 2023-11-17
>
> We thank the reviewer for their comments. We reply to the weaknesses and questions.
>
> **Q. Connection between geometric modularity and density-based clustering.**
>
> Thanks for raising this point. We will clarify it in the final version. The main reason for introducing this notion in metric spaces is that input to density-based clustering algorithms is often points in a vector space. We focus on DBSCAN because (i) we were able to give a theoretical explanation regarding the effectiveness of using geometric modularity, (ii) the empirical performance is also very good. DBSCAN is also one of the most widely used clustering algorithms. For $k$-means, etc. it is not clear how we would tune the hyperparameter $\rho$ as we mention in our reply to Reviewer WBYe. Exploring further applications of this notion is an interesting direction for future work.
>
> **Q. Experiments for reproducibility.**
>
> Thanks for the suggestion. We will add an Appendix with full details of all the parameters. We will also make the code available before publication.
>
> **Q. HDBSCAN.**
>
> We will report this experiments in the final paper. And also post results on this discussion thread as soon as they are available.

---

### Official Review · Reviewer_WBYe · 2023-10-31

**Soundness:** 3 good
**Presentation:** 3 good
**Contribution:** 1 poor
**Rating:** 3
**Confidence:** 3

**Summary:**

This manuscript proposes using geometric modularity as a quality to improve the performance of DB-SCAN. In fact, the proposed algorithm outputs a clustering with the highest geometric modularity. It is also claimed that the geometric modularity has a positive relationship with adjusted mutual information. The empirical study is done with quite a number of datasets and compared with other two methods.

**Strengths:**

1. The paper is __in general__ well-organized and conveys the contents with clarity.
2. The experiments are analyzed carefully, and the results show good performance of the proposed method.
3. Geometric modularity is known as a 'metric' to evaluate the clustering results. It is good to formally build this connection.
4. It's also good to remind people of the linear computation of the geometric modularity.

**Weaknesses:**

1. Section 4.1 starts with "Our algorithm is outlined in Section 2". However I am not able to find a procedure such as Alg 1. It is not a big issue but for better clarity I think there should be one.

2. I have three main concerns: (1) limited novelty, which leads to weak theoretical analysis; (2) overclaimed contribution; (3) parameter choice of $\rho$.
- (1) Geometric modularity is used frequently for evaluation if there's no label at all (in contrast to pseudo-unsupervised), it is already a routine to examine clustering results. Therefore introducing an algorithm simply maximizing it does not shed new light. This also leads to a weak theoretical analysis which is basically proving what is assumed. I understand the authors want a "reasonable" datasets. But the point of beyond-worst-case analysis, per my understanding, is to avoid pathological instances but not to design properties of datasets tailored to what we want to prove. In definition 3.1, for example if one item is removed or relaxed, the proof breaks down.

- (2) I agree with the claim that "The unsupervised geometric modularity tracks a supervised measure AMI". But the positive relationship between this two measure is also known to the community, especially in the lens of density-based clustering. I am not sure if any previous work formally states this, but in this paper it is still shown by experiments. But anyway I can stay open to discussion.

- (3) If I am understanding correctly, the motivation is to get rid of the parameter choice on $\varepsilon$, then what is the point if we have to choose $\rho$ again? Indeed, theorem 3.2 gives a range, but it is subject to other parameters given by the dataset, and according to section 4, it is selected in [0.5, 1].

3. The experiments include quite a number of datasets but not enough clustering methods. For example, SOM, HDBSCAN, or even $k$-means with the best $k$.

**Questions:**

- I wonder if computing the modularity in linear time is one of the contributions of this work? If not, is it folklore (just a practical heuristic) or proposed by previous work?

- I do not see a straightforward break-down if we apply the geometric modularity to other clustering methods, even $k$-means/median/center. Is there any foreseeable problem?

---

> ### Author Response · Authors · 2023-11-17
>
> We thank the reviewer for their comments. We reply to the weaknesses and questions.
>
> **Q. No algorithm description.**
>
> We thought that the algorithm as described in the text was sufficient, but thanks for pointing this out, we will add it in the new version.
>
> **Q. Geometric modularity frequenty used.**
>
> We are very confused by the statement that geometric modularity is frequently used. We are not aware of any work mentioning “geometric modularity”. We acknowledge that there are various works on graph modularity but one of the novelties of this work is to introduce a notion of modularity for (pseudo)metric spaces.
>
> It seems that there is a confusion here: the fact that the input exhibits clusters does not mean that the algorithm can find them. The assumptions used here only state that there is an underlying clustering and not that this clustering is tailored to the algorithm since similar assumptions have appeared in the literature before. In fact, such an analysis is lacking for most algorithms and, for example, DBSCAN with poor parameter choices would fail to recover the clusters.
>
> **Q. Agree with claim Geometric Modularity tracks AMI", but this is known to the community.**
>
> We are not aware of any work connecting geometric modularity with AMI (also because we are not aware of any other work mentioning the quantity “geometric modularity” that we introduce here). If the reviewer could point us to these works, we could reference them in our paper.
>
> **Q. Get rid of parameter choice, what is the point?**
>
> The main point here is that there is no such thing as getting rid of parameters. The only way to do this is to set them to default values and show that they are good, or to show that they can be auto-tuned using a data-dependent fashion. We take the latter approach. We believe it is not straightforward to choose $\epsilon$ in a data-dependent fashion, but picking $\rho$ is easier as outlined in Section 4. As $\rho$ can be auto-tuned, once $\rho$ is fixed we use geometric modularity can be used to pick $\epsilon$.
>
> **Q. More experiments with HDBSCAN, $k$-means.**
>
> Thanks for the suggestion we will add experiments with  HDBSCAN. For $k$-means we are less sure what the reviewer has in mind. Selecting $k$ is indeed the part hard of the problem (in practice) and so for a fair comparison that will require us comparing heuristics that are very different from those for density based clustering, which is our focus in this paper.
>
> We will post results with HDBSCAN on this discussion forum as soon as they are available.
>
> **Q. Computing modularity in linear time as a contribution.**
>
> Graph Modularity would require a computational time proportional to the number of edges of the graph (which is at least as large as the number of vertices). Our new geometric modularity framework takes a more global vision of the dataset (by looking at all pairwise distances) and that can be done in time linear in the number of data elements. This is indeed one of the contributions, and while the proof itself is simple, we believe this is what makes the approach attractive in practice.
>
> **Q. I do not see a straightforward break-down if we apply the geometric modularity to other clustering methods, even k-means/median/center. Is there any foreseeable problem?**
>
> This is indeed an interesting research direction to explore. However, as mentioned above picking $k$ itself is not usually an easy problem, and unlike in the case of density-based algorithms it is not clear how we would pick $\rho$. Although Geometric modularity with a fixed value of $\rho$, e.g. 0.8 does work well most of the time, to obtain better results, we do need to be able to tune $\rho$.

---

### Official Review · Reviewer_KqpR · 2023-10-31

**Soundness:** 2 fair
**Presentation:** 1 poor
**Contribution:** 1 poor
**Rating:** 3
**Confidence:** 4

**Summary:**

The paper describes a quality measure for clustering called "geometric modularity",
which is used to tune DBSCAN's radius-parameter (called DBSCAN-mod).
The authors show that the ground-truth labels of well-seperable idealized data maximizes the geometric modularity.
They found that DBSCAN-mod can achieve the ground-truth under these idealized circumstances.
Experimentally, the authors discovered that DBSCAN-mod often achieves a higher adjusted mutual information (AMI) than the OPTICS algorithm and observe.

**Strengths:**

Using the graph-theoretical modularity measure for community structures, to assess clustering is a reasonable proposition.
The paper considers a large body of 15 real-world benchmark datasets.
The paper discusses the impact of outliers on the result.

**Weaknesses:**

1. The proposed geometric modularity seems identical to Newman's weighted modularity [2004, https://journals.aps.org/pre/abstract/10.1103/PhysRevE.70.056131],
but this link to a well-established variant has not been made by this paper.

2. The experimental evaluation lacks in depth.
The claim that DPC did not result 'in any meaningful results' is not supported by facts.
The authors only compare against OPTICS.
Widely-used modularity-optimizing algorithms (albeit parametric and on weighted adjacency matrices), are not part of the experimental evaluation.
There is no experiment showcasing the limitations of the tuning procedure under noise or high-dimensional data.
The computational complexity analysis is not particularly insightful and of practical use.

3. The empirical results lack context.
That is, the paper does not compare geometric modularity to other `competing' ("internal"/"unsupervised") clustering-quality measures for tuning DBSCAN,
including the classical Silhouette coefficient, Davies–Bouldin index, or Dunn index (under CV or model selection criteria).

4. The writing is unclear, not technical, and includes weasel-words, hyperbole, and unspecific adjectives "few papers", "some", "tracks AMI incredibly well", "better output quality", or "not much has been done".
The paper does not properly motivate the usage of AMI well.

5. The paper does not properly describe their local-search post-processing heuristic, which had a profound impact on the performance, thus inhibiting the reproducibility.

6. The paper does not include a Reproducibility Statement and the submission does not include a Reproducibility Package.

**Questions:**

I don't understand the formal argument and the implication on the identifiability of the optimal solution, of your suggestion to smooth-over the hard-to-optimize (rugged?) solution landscape using isotonic regression. Could you expand on this?

---

> ### Author Response · Authors · 2023-11-17
>
> We thank the reviewer for their comments. We reply to the weaknesses and questions.
>
> **Q. Relation to Newman Modularity.**
>
> Geometric modularity is a modularity framework for Euclidean spaces, it deals with distances and not similarity edges. This is a fundamental difference between the two definitions. In particular, the definition in [Newman](https://journals.aps.org/pre/abstract/10.1103/PhysRevE.70.056131) would have limited meaning in Euclidean space and would not have the properties that we show for our definition. We note that we have cited Newman’s paper on graph modularity and we will add a citation to this one (thanks for the pointer).
>
> **Q. Experimental Results.**
>
> DPC always reports one cluster in the experiments we have performed (and simply doesn't run in reasonable time for larger datasets), the bound on AMI follows directly. Although it is possible that hyperparameters of DPC could be tuned, as our key contribution is better auto-tuning of DBSCAN hyperparameters, we believe it is fair that we compare to other available methods out of the box that also claim to autotune hyperparameters.
>
>
> The fact that the geometric modularity (for squared Euclidean distance) can be computed in linear time is actually key to the implementation. Computing other measures (as the one suggested below) requires a time quadratic in the number of input points, and so are not scalable. The geometric modularity measure is the only measure among the ones listed above that is scalable.
>
> **Q. Writing. Motivation for AMI.**
>
> We apologize if the writing appeared as lacking rigor. We are working on an improved version that we will soon post. AMI is one of the most standard ways of measuring cluster quality — proposed as an improvement to NMI which often favors clusters with outliers.  See for example [[1]](https://proceedings.mlr.press/v139/gosgens21a.html).
>
> **Q. Description of local search heuristic.**
>
> Thank you for your comment, we will add this to the new version we will post. But in short, it is the standard local search approach, tentatively move one point from one cluster (picked at random) to another (random) cluster - accept the move if the geometric modularity increases, reject otherwise. Stop if no accepting move is found for a certain preset number of attempts.
>
> **Q. Reproducibility Statement.**
>
> We will also include this in the new version we post.
>
> **Q. Isotonic Regression.**
>
> The only place we suggest using isotonic regression is to autoselect $\rho$. While we expect the #clusters obtained to be a decreasing function of $\rho$, this is not a formal statement as DBSCAN clusters may exhibit different behavior from expectation (this was not the case in our experiments). Since the way we select $\rho$ is to pick the most stable part of the curve for #clusters v $\rho$ we recommend making the curve monotonic (if needed to remove outliers) before picking $\rho$. We will clarify this in the Appendix in the revised version of the paper with some plots and more explanation.
>
> [[1]](https://proceedings.mlr.press/v139/gosgens21a.html) Systematic Analysis of Cluster Similarity Indices: How to Validate Validation Measures. Marijn M. Gösgens, Alexey Tikhonov, Liudmila Prokhorenkova. Proceedings of the 38th International Conference on Machine Learning. PMLR 139:3799-3808, 2021.

---

### Official Review · Reviewer_AWS2 · 2023-11-05

**Soundness:** 3 good
**Presentation:** 3 good
**Contribution:** 3 good
**Rating:** 6
**Confidence:** 4

**Summary:**

This paper propose a new measurement of clustering quality called geometric modularity (GM), which extends the previous definition of modularity on graph data structure to vector-valued data samples. Given a clustering result, GM essentially measures the difference between the data density between intra-cluster and inter-cluster, and thus a higher geometric modularity indicates a stronger evidence of cluster patterns. The authors shows the GM can be evaluated in linear time, and use GM to guide the parameter tuning of commong clustering methods such as DB-SCAN. Experiments demonstrates that the output from DB-SCAN tuned by GM usually yields better cluster quality then existing approach such as OPTICS.

**Strengths:**

A list of strengths:
- This paper is well-written and easy to follow
- The idea of extending the modularity to vector-valued data is interesting and seems to be effective. The analysis on the linear runtime is important to make the geometric modularity computationall suitable for pratical use.
- The theoretical analysis on the exact recovery by maximizing the geometric modularity is a good plus, and appers to be technically correct.
- The experiments is comprehensive and demonstrates a significant improvement compared to existing methods.

**Weaknesses:**

- Given that the modularity ifself can be used for community detection in network data, and various approaches exist for clustering by maximzing the modularity, such as those mentioned in [1]. I think it is natural to ask if we can also apply similar approahces to the geometric modularity here, without using any auxiliiary tools such as DB-SCAN. If there are any critical challanges e.g. computational complexity, it might be worthy to point these out, and such challenges can be also good motivations for instead applying geometric modularity to DB-SCAN. Currently, it seems like there is a lack of such discussion.
- I was a little bit confused about the originarity of the geometric modularity, as it is defined very similar to Eq.(1.1) in [2] which is also mentioned by the authors. Therefore, I think in prior to introducing the geometric modularity, it might be better to give a more detailed introducing fisrt on the modularity on unweighted graph, such as  Eq.(1) in [1], then its generalized version on weighted graph, e.g. Eq.(1.1) in [2]. As a result, the geometric modularity can be better motivated and their differences (I think the biggest difference is on the extra parameter $\rho$) is more clearly demonstrated.
- In the experimental results, the performance of OPTICS is dramatically poor compared to the DB-SCAN with geometric modularity. However, given that the high similarity between OPTICS and DB-SCAN, I would not expect such a huge performance gap exists, and thus I am very curious on the root causes, which could fall onto the following two categories from my opinions:
    1. The methodology of OPTICS essentially fails in these datasets, no matter how we choose the hyper parameters
    2. The result of OPTICS is sensitive to the hyper-parameters, and the parameters chosen in the experiments is still far from the optimal one.

- Currently, the answer is still not clear to me eventhough the result in Section A.3 might shed some light. Finding the root cause is important as it provides the more evidence on why we should use geometric modularity rather than OPTICS. If the reason is on the methodology of OPTICS, I would expect to see some plots on the ordered list of reachable distances generated by OPTICS, and see if there is indeed no clear clustering pattern no matter how the steepness is chosen. If the reason is on the hyper-parameters, maybe a wider range of $\xi$ should be verified in Section A.3


[1] Newman, Mark EJ. "Modularity and community structure in networks." Proceedings of the national academy of sciences 103.23 (2006): 8577-8582.

[2] Arenas, Alex, Alberto Fernandez, and Sergio Gomez. "Analysis of the structure of complex networks at different resolution levels." New journal of physics 10.5 (2008): 053039.

**Questions:**

- Could the author provide some discussion on why we don't consider clustering by geometric modularity itself, and what is the possible challenges on doing this?
- Could the author clarify the originarity of the geometric modularity, and state the main difference between the previous works?
- Could the author explain the siginificant performance gap between OPTICS and DB-SCAN, and see what is the root cause of that?

Of course, please let me know if I missed anything. I would be very glad to raise my score if these questions can be properly addressed.

---

> ### Author Response · Authors · 2023-11-17
>
> We thank the reviewer for their comments. We reply to the weaknesses/questions, which we believe are related.
>
> **Q: Why not geometric modularity itself?**
>
> Thanks for pointing this out. Optimizing geometric modularity directly is an interesting open problem. Optimizing directly is NP-hard (same reason as the original modularity) and we do not think that classic approaches such as local search would lead to any formal result in this space. We focused on a new algorithm based on DBSCAN because it had interesting theoretical properties (as shown in our paper) and also because the experimental results were quite interesting. We will expand the discussion in the final version of the paper.
>
> **Q. Clarification about geometric modularity and differences with prior work.**
>
> Thanks for pointing this out. A key difference between the two modularities is that one deals with distances (our case) and the other with similarity. Although the definition of geometric modularity itself does not require $d$ to be a metric, the theoretical analysis we present does as it uses the triangle inequality (our theoretical analysis also works with somewhat weaker variants of triangle inequality, but some constraints are required). This is why we cannot directly adopt a version of modularity defined over weighted graphs; though the definition itself is very similar and is indeed inspired by work in graph modularity.
>
> **Q. Performance Gap between OPTICS and DBSCAN.**
>
> Although OPTICS does give a way to automatically perform hyperparameter selection on DBSCAN, it does not in fact consider all possible (or even a large range of possible) values of $\epsilon$ and then perform hyperparameter autotuning. Instead, it uses a clever heuristic to order the points according to distances, and looks for steep changes in these distances as a means to identify when a new cluster has been discovered. This parameter $\xi$ is what controls this and its effect is also presented in our additional experiments. This approach works quite well in very low dimensions, but starts to suffer in even modest dimensions.

---

### Meta-Review · Area_Chair_upnV · 2023-12-10

**Metareview:**

Many reviewers had rather critical remarks, such as unclear novelty over the idea of Newman’s weighted modularity,  limited scope and depth of the experimental evaluation and clarity of writing, and during the discussion period it became clear, that even the reviewers who assigned a more positive score shared these concerns to some degree. In my opinion, the rebuttal could not fully address al these issues, and there are still several open questions regarding the novelty and depth of the contribution. Therefore, I recommend rejection.

**Justification For Why Not Higher Score:**

Too many open questions about the novelty and depth of the contribution.

**Justification For Why Not Lower Score:**

N/A

---

### Decision · Program_Chairs · 2024-01-16

Reject